# Examination of CIs in health and medical journals from 1976 to 2019: an observational study

Adrian Gerard Barnett  ,[1] Jonathan D Wren[2,3]

[1]Institute of Health and Biomedical Innovation, Queensland University of Technology, Kelvin Grove, Queensland, Australia
[2]Arthritis and Clinical Immunology Research Program, Division of Genomics and Data Sciences, Oklahoma Medical Research Foundation, Oklahoma City, Oklahoma, USA
[3]Department of Biochemistry and Molecular Biology, University of Oklahoma Health Sciences Center, Oklahoma City, Oklahoma, USA

**Correspondence to**
Dr Adrian Gerard Barnett;
a.barnett@qut.edu.au

## ABSTRACT

**Objectives** Previous research has shown clear biases in the distribution of published p values, with an excess below the 0.05 threshold due to a combination of p-hacking and publication bias. We aimed to examine the bias for statistical significance using published confidence intervals.

**Design** Observational study.

**Setting** Papers published in *Medline* since 1976.

**Participants** Over 968 000 confidence intervals extracted from abstracts and over 350 000 intervals extracted from the full-text.

**Outcome measures** Cumulative distributions of lower and upper confidence interval limits for ratio estimates.

**Results** We found an excess of statistically significant results with a glut of lower intervals just above one and upper intervals just below 1. These excesses have not improved in recent years. The excesses did not appear in a set of over 100 000 confidence intervals that were not subject to p-hacking or publication bias.

**Conclusions** The huge excesses of published confidence intervals that are just below the statistically significant threshold are not statistically plausible. Large improvements in research practice are needed to provide more results that better reflect the truth.

## INTRODUCTION

Every health and medical researcher is aware of the importance of statistically significant results, generally meaning a p value less than 0.05. A study with statistically significant results is easier to publish in a journal[1] and statistical significance can be mistaken for study validity and importance.[2] This can lead some researchers to strive for statistically significant results by reanalysing data to get a p value under 0.05, known as 'p-hacking'.[3 4] This striving for statistical significance is not always overt and can occur due to researchers making seemingly sensible scientific decisions.[5] It can also motivate researchers to carefully vet which hypotheses they pursue, selecting those they deem likely to yield significant results. For example, around 75% of scientific effort goes towards the 10% of genes already best characterised, suggesting widespread risk-aversion.[6]

### Strengths and limitations of this study

► This is the first study to examine the bias towards statistical significance using confidence intervals instead of p values.
► We used a very large sample of confidence intervals from both abstracts and full-texts.
► We used a simple graphical summary and did not develop a new test to detect p-hacking or examine differences in papers around the statistically significant threshold.
► We have taken a broad look at the combined evidence and have not examined individual studies to show that they were biased.

Journals are also somewhat responsible for the bias towards statistically significant results. Journals depend on readership, and journal editors know that readers tend to be more interested in striking results.

The combined 'significance seeking' behaviour of researchers and journals has created a statistically implausible excess of published p values just below the 0.05 threshold.[1 7 8] This warped evidence undermines the purpose of evidence-based practice, as 'negative' studies too often go unpublished and some published effect estimates are too large because p-hacking inflates effect sizes (by conditioning on statistically significant results). This biased evidence can harm the public's health when policy decisions are made using incomplete or overly optimistic evidence.

Examining published p values using the 'p-curve' distribution can be used to indicate that p-hacking has occurred.[3] In this paper, we show that the same excess occurs in confidence intervals, which are a recommended alternative to p values for presenting results and have been advocated as a way to avoid 'bright-line' thinking at the 0.05 p value threshold.[9–11] We therefore take an alternative look at the already illustrated bias for statistical significance. However, we think

this is worthwhile replication because this bias is a crucial problem for evidence-based medicine. Also using confidence intervals may lead to a wider understanding of the bias because researchers often misunderstand p values.[12] We also show that the bias has not abated in recent years.

As a reminder, a p value is the probability, under a specified model, of observing a test statistic as or more extreme than that observed in the data. A 95% confidence interval is a range that should contain the true value on 95% of occasions if the data generating process could be repeated many times. Most confidence intervals are given as 95% intervals, which correspond to a 0.05 p value threshold.

## METHODS

We extracted ratio confidence intervals from the abstracts and full-texts of journals using regular expressions; for full details, see Georgescu and Wren.[8] The text-mining algorithm was designed to recognise the typical ways in which ratio estimates (eg, odds ratios, hazard ratios and risk ratios) and confidence intervals are presented. The results were independently validated using a separate text-mining algorithm developed by the first author (AGB). Random checks were made on approximately 5000 abstracts and 500 full texts, as well as other checks on unusual results, for example, extremely wide confidence intervals. The complete codes and data are available at https://github.com/jdwren/ASEC and https://github.com/agbarnett/intervals.

We excluded the 0.1% of lower intervals that were zero, assuming this was an error by the authors. We excluded the 0.7% of abstract intervals and 0.9% of full-text intervals where the mean was not within the confidence interval, as this meant either the interval or mean (or both) was incorrect.

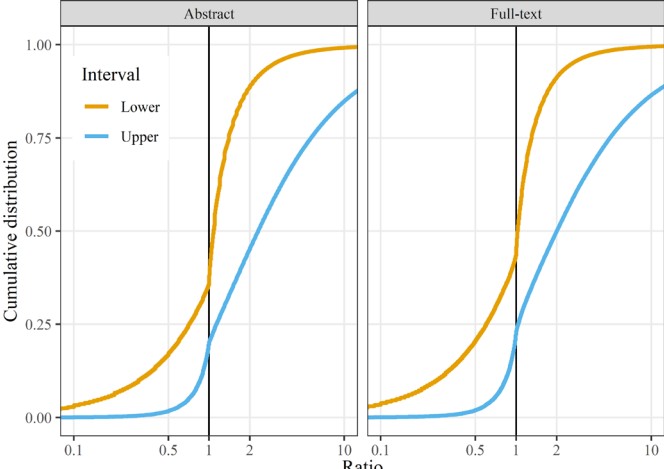

**Figure 1** Empirical cumulative distributions for ratio CIs from *Medline* abstracts and full papers. To be statistically significant, lower intervals need to be above 1 and upper intervals need to be under 1 (vertical lines). The x-axes are restricted to ratios between 0.25 and 4.

We used a second set of published estimates from health research based on thousands of analyses with no p-hacking or publication bias.[13] Using four large databases of insurance claims, the study examined all possible pairs of 17 treatments for depression using 22 outcomes. Risks were estimated using survival analysis, giving hazard ratios. Treatment–Outcome pairs were categorised as 'negative' where there was no evidence in the literature of any association. 'Positive' associations were created by simulating an increased risk with a hazard ratio of 1.5. Details are in the paper by Schuemie *et al*;[13] the key point for our analysis is that we have a large sample of ratio confidence intervals from Treatment–Outcome pairs that are not subject to any 'significance seeking' by researchers or journals.

### Statistical methods

To summarise the intervals, we plotted the cumulative empirical distribution of the lower and upper confidence interval limits. We used the cumulative distribution because it highlights a change in the distribution, which is our key interest, and it does not require any user-defined tuning parameters. Histograms require the selection of a binwidth and density plots require user-defined smoothing parameters, which are subjective and will create different impressions depending on the choice made. We include histograms in the online supplementary file using a binwidth of 0.1, but our results focus on the cumulative distributions.

Our hypothesis was that there would be a large change in the intervals near 1, which is the commonly used null hypothesis of no difference on a ratio scale. A lower confidence interval limit for a ratio estimate that is greater than one would mean a statistically significant result, as would an upper interval limit below 1. To look for changes over time, we plotted separate cumulative densities in 5-year periods.

We used R V.3.6.0 for all analyses.[14]

### Patient and public involvement

No patients or members of the public were involved in this study.

## RESULTS

Our sample had over 968 000 intervals extracted from *Medline* abstracts from over 5900 journals, and over 350 000 intervals extracted from the full-text from over 2700 journals.

For 11% of intervals, the level of the confidence interval could not be determined, most often because it was not given. Where an interval could be determined, 99.7% were 95% confidence intervals, which corresponds to the commonly used p value threshold of 0.05.

We found a clear and sudden change in the cumulative distribution of both lower and upper interval limits at the statistically significant threshold of 1 (figure 1). There was a steep increase in the number of lower interval limits

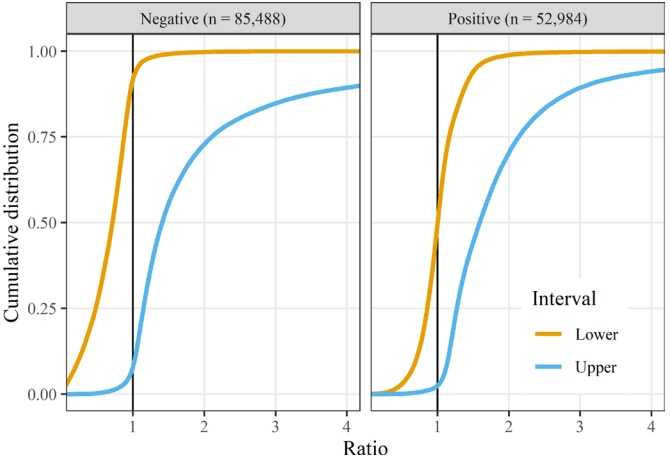

**Figure 2** Empirical cumulative distributions for ratio CI limits from a sample with no p-hacking or publication bias. 'Negative results were for associations with no known evidence of an association and 'positive' studies were simulated using an increased risk. The x-axes are restricted to ratios between 0.25 and 4.

just above 1, meaning they were just above the threshold for statistical significance. Similarly, there was a steep increase in the number of upper interval limits just below 1 and so just inside the statistically significant threshold. The discontinuity at 1 appears slightly stronger (less smooth) in the abstract than the full-text and there are noticeably more lower intervals below 1 in the full-text, which suggests that the biases are stronger in the abstract than the full-text.

If there was no p-hacking and publication bias, the cumulative distributions would be smooth S-shapes with no discontinuities, and this is confirmed in figure 2 using the data without these biases.

The bias for statistically significant results in abstracts has persisted over time as shown by the 5-year plots in figure 3. The plots also show the reduced use of rounded

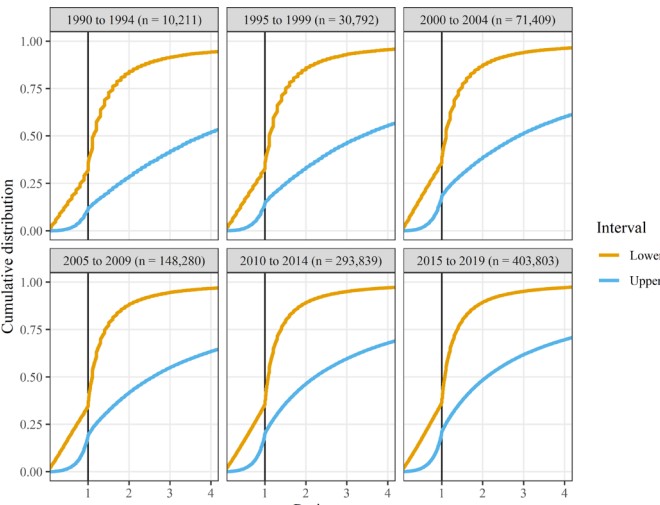

**Figure 3** Empirical cumulative distributions for ratio CIs from *Medline* abstracts in 5-year intervals. The x-axes are restricted to ratios between 0.25 and 4.

intervals over time, as the cumulative densities are smoother in later years.

The online supplementary file contains sensitivity analyses that use wider and narrow ratio axes, show histograms instead of cumulative densities and show plots of the cumulative densities over time for the full-text. All the plots show the same striking changes in the number of intervals near 1.

## DISCUSSION

The strong bias for statistically significant results is clearly visible in the confidence intervals of a large sample of abstracts and full-texts from health and medical journals and has persisted for decades. This bias undermines the evidence base and impairs good decision making by giving a false picture of the effectiveness of treatments and procedures. Such biased evidence can harm patients and the public when it gets translated into policy and practice.

Confidence intervals have been promoted as a way of avoiding the simplistic bright-line thinking that often accompanies p values. However, our results show that even though people are frequently reporting confidence intervals, many are still focusing on statistical significance. A push towards the greater use of confidence intervals in journals in place of p values may therefore not solve the ongoing problem of bright-line thinking.[15]

Large improvements in research practice are needed to improve the quality of evidence. Study results should be presented and published regardless of their statistical significance. This could be achieved by a wider uptake of study preregistration and analysis plans,[16] which reduces the potential for researchers to change their hypothesis or primary outcome after seeing the data.

Statistical significance should no longer be used as a tool to screen what results are published and the evidence base would be in a better state if significance were given far less prominence.[17] The prominence given to statistical significance is ironic given the widespread misunderstandings of its true meaning.[2 12]

### Limitations

We did not aim to develop a new statistical test for p-hacking or numerically compare the extent of p-hacking in abstracts versus full-texts or our sample of the published literature versus the sample with no p-hacking. Instead our aim was to present some simple plots that highlight the extent of the problem.

We included all confidence intervals, but we anticipate the biases would be even more striking if we were able to restrict the sample to the primary outcomes of interest.[18] This is because the statistical significance of the primary outcomes dictate the 'success' or 'failure' of the study, and less high-profile comparisons (eg, between groups at baseline) are less likely to be p-hacked.

We have taken a broad look at the combined evidence and have not examined individual studies to show that

they were biased, nor did we estimate the size of the bias. Instead, we aimed to show the extent of the bias across the literature using a simple graphical method. To our knowledge, this is the first paper to examine the problem using confidence intervals and uses one of the largest sample sizes including results from both abstracts and full-texts. For us, the graphs are a warning of the urgent need for action to improve research practices.

**Contributors** AGB and JW conceived the idea. JW wrote the programs to extract the data. AGB wrote the analyses code and the first draft of the paper with comments from JW. AGB is responsible for the overall content and is the guarantor.

**Funding** AGB was supported by Queensland University of Technology and the National Health and Medical Research Council grant number APP1117784.

**Competing interests** None declared.

**Patient consent for publication** Not required.

**Provenance and peer review** Not commissioned; externally peer reviewed.

**Data availability statement** Data are available in a public, open access repository.

**ORCID iD**
Adrian Gerard Barnett http://orcid.org/0000-0001-6339-0374

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
