## [Reviewer comments · BMJ Open]

ARTICLE DETAILS

TITLE (PROVISIONAL)	An examination of confidence intervals in health and medical journals from 1976 to 2019: an observational study
AUTHORS	Barnett, Adrian; Wren, Jonathan

VERSION 1 – REVIEW

REVIEWER	Charles Poole University of North Carolina United States of America
REVIEW RETURNED	17-Jul-2019

GENERAL COMMENTS	General comments 1. The authors have done many research fields a good service by refining the examination of distributions of two-sided null p-values with specific attention to a bolus just below 0.05 (which I'll call the "barely significant bolus" BSB). A BSB can provide evidence of some, but not all, behaviors that impart selection biases to published literatures. Here, the authors extend this valuable work to BSBs of lower 95% confidence limits just above the null value and BSBs of upper 95% confidence limits just below the null value.2. When authors bother to define the term "p-hacking," they do so in different ways. The authors give a rather narrow definition (p.3) that is the same as in reference 3: re-analyzing the same association in the same set of data in different ways in search of statistical significance (traditionally called "cooking"). Reference 4, however, gives a broader definition that includes not just cooking but varying the inclusion criteria for the analytic data set ("trimming") and extending the study in some way to collect more new data (e.g., by changing the "stopping rule"). Some authors might include even more behaviors under the "p-hacking" rubric, such as fraudulent fabrication or alteration of data ("fudging"). Any of these behaviors can produce a BSB. Hence, if the authors want to say that a BSB can identify or "detect p-hacking" (p.2 et seq.), I would recommend defining the term so that it includes every
--

	behavior that can produce a BSB and excludes every behavior that can't. 3. In previous work, the authors have argued persuasively that, though publication bias and p-hacking can produce an excess of statistically significant results in the published literature, only p-hacking can produce a BSB. In this paper, that distinction is not drawn quite as sharply or explicitly as it might be. I would encourage the authors to bring it out into the open and summarize the argument. 4. Experienced data analysts, among whom I would have to count myself, have reported anecdotally that, short of fudging, it's hard to make the null p-value change a great deal by the behaviors that can produce a BSB. If they are right, we might expect a shortfall in the p-value and confidence limit distributions in the region just barely on the statistically nonsignificant side of the divide. (If they have addressed this possibility in previous work, I apologize for not being familiar with it.) Specific comments 5. (Supplement) I don't think the STROBE checklist is appropriate here. It's for individual studies, not reviews. 6. (p.3) Please give the specific page numbers on which reference 2, a book, gives or refers to empirical evidence on the frequency with which statistically significant results are perceived as being more important and valid. Of particular interest would be any empirical evidence, cited in reference 2 or elsewhere, showing that researchers equate statistical significance with negligible bias (i.e., with validity). 7. (p.3) Is carefully vetting which hypotheses to pursue, selecting the ones for which a study has the greatest statistical power and therefore gives the most precise estimates, a bad thing? 8. (p.3) Please provide references that distinguish the role of journals from the roles of researchers, employers, and funding sources in publication bias, p-hacking, or both. 9. (p.3) With regard to the assertion that journals "are also somewhat responsible," I wonder if the authors would address, or at least mention, the possibility of the following intergenerational conditioning effect: Suppose journals condition generation after generation of researchers to expect an uphill battle to get their statistically nonsignificant results published. Then the journals become enlightened, change their policies without fanfare, and quietly abandon their
--	---

	preference for statistically significant results. From that point on, it will misleadingly appear as though the fault lies with the researchers, whereas the underlying cause is still the past censorship by the journals.
--	--

REVIEWER	Patrizio Tressoldi Dipartimento di Psicologia Generale - Università di Padova, ITALY
REVIEW RETURNED	24-Jul-2019

GENERAL COMMENTS	This is a new contribution to the problem of publication bias consequent to the incentives to publish studies with only statistically significant results. The authors clearly show that in the studies examined, there is a clear "confidence intervals - hacking". I have some minor comments to add.  - both in the abstract and in the main text add the time intervals of the retrieved studies (from 1976 to ...). Furthermore, in the main text add the description of the search method; - on pag. 3, line 50, it could be added "...under a specified model, usually a null hypothesis testing, ... - in the Discussion, add a comment whether the observed CI - hacking may be prevalently due to errors in CI estimation or in intentional rounding or adjustments;
---

REVIEWER	Nicole Lazar, Professor University of Georgia, USA
REVIEW RETURNED	09-Sep-2019

GENERAL COMMENTS	I'd like to see a clearer explanation of p-hacking (page 3, top). The authors write that "reanalysing data to get a p-value under 0.05" is known as p-hacking. This is a bit misleading, as one could infer that any secondary analyses are p-hacking. But prespecified secondary analysis would not be considered p-hacking typically. It should be made more clear that p-hacking occurs when the data are subjected to unplanned analysis (e.g. stratifying of units in the study according to their characteristics) in the search for "statistical significance" (typically $p < 0.05$). Likewise, the statements about the part that journals play could be clarified. Readers who are not as immersed in this literature and debate may not understand exactly the connection between the "statistically significant" and the "striking" result - elaboration here would be useful. The authors want to set the stage for the problem, but I find the discussion too general and likely to mislead readers who aren't already following the literature on this topic. I'd consider dropping the reference to the p-curve. It plays no further role in the work, and is not directly relevant. Instead, include references to papers that have investigated the excess of reported p-values close to the traditional cutoff points (0.01, 0.05, 0.1 are typical). I don't think it's precisely correct to say that confidence intervals have been "advocated" as an "alternative to p-values" for presenting
--

	results, on account of avoiding bright-line thresholds. After all, a 95% CI gives a type of bright-line of its own (why not 90%? or 87%? etc.) Rather, I think the claim is that a confidence interval provides more information, for the same “cutoff value” about what values of the parameter are compatible with the data. It’s a perhaps subtle distinction, but an important one. CIs can be abused just as easily, as this paper demonstrates! While it’s no doubt true that researchers often misunderstand p-values, are confidence intervals any better understood? I’m not convinced and this does not therefore seem like a strong basis for studying confidence intervals in the way proposed. One could be more direct about it! In the Methods section, what is meant by “ratio confidence intervals”? (this is imprecise language) Ratio of what to what? It’s made clearer in the next sentence but it might be better to lead with some context. Why are CIs of ratio estimates of particular interest? Presumably these are the most prevalent in the health research literature under study here? This should be clarified. Also it would be helpful for readers to provide more information about the text mining algorithm. The code alone may not be sufficient without some additional background. Or provide reference to a relevant supporting document. This may be in the Georgescu and Wren paper, but then what is the second text mining algorithm developed by the first author? There is not enough detail about this key aspect of the methods. On page 5, under results, the phrasing about the upper interval limits just below 1 (“and so just inside the statistically significant threshold”) is not clear. Regarding Figure S.3, the authors note that the steps in the ratios that are evident in the “zoomed in” plots are not simply due to rounding, since the discontinuity at 1 is clearly visible. But it does look like the steps at 1.1 and 1.2 (but, interestingly not at 0.9!) are of the same magnitude as the jump at 1. This is true in both the left and right panels (abstract and full-text, respectively), though somewhat more obvious in the former. Indeed the discontinuity at 1.1 for the intervals reported in the abstracts looks to be bigger than the jump at 1. I think this needs further exploration before one can claim that the cutoff of 1 has special behavior/bias.
--	---

REVIEWER	Paul Gustafson University of British Columbia, Canada
REVIEW RETURNED	15-Sep-2019

GENERAL COMMENTS	I read this work with interest, and I am all around positively disposed to it. I have some comments and suggestions though. p. 2: “are not statistically plausible” – could be misread as in indictment of statistical methods, when in fact the charge is levelled at research practice generally p. 3: Worth mentioning also funding agency practices and university promotion practice as incenting “significance seeking”?
---

	p. 3, last para: little bit awkward to discuss correspondence of testing and interval estimation without some mention of “null” hypothesis. p. 4: “mean was not within the CI” -> “point estimate was not within the CI”? (No real sense in which any of these estimates of *ratios* are means.) p. 4: “highlights a change in the distribution” is, to my mind, awkward wording. A cdf is by definition an increasing function, so it is “changing” at every point along the abscissa. p. 5: smooth..., with no discontinuities. To my mind (and to anyone interpreting with some level of math formality), “discontinuity” is not the right word. The cdf is continuous at $x=1$, but you are calling attention to a discontinuity in its derivative (the density function). I’m not sure what word is most appropriate, but a “bend” (versus not), would be a candidate. p. 6: “should be published regardless of their stat significance” ... to my mind, this is way too simplistic. This change, in isolation, would lead to journals as dumping grounds. Investigators would be incited to perform underpowered studies (even more so than is now the case), and it also punts on the question of going after novel/important/breakthrough hypotheses. I think the work could be better anchored to the existing literature. In particular, there aren’t citations to what are probably the two biggest recent “splashes” in this space – the special issue of Am. Stat. and the Nature commentary.
--	--

REVIEWER	Theis Lange Section of Biostatistics, University of Copenhagen
REVIEW RETURNED	16-Sep-2019

GENERAL COMMENTS	The paper presents an interesting work on possible publication bias can be "seen" by looking at limits of confidence intervals. The short answer they find is: Yes it can. By far the most impressive piece of work is the collection and coding of close to 1M confidence intervals from MEDLINE. The methods described for achieving this looks convincing, but it is of course impossible to review the precision of this part of the procedure. Based on this impressive data set the authors conduct some relative simple descriptive graphical analyses which indicates that there is a publication bias. Comments: 1) The section on how control CIs are obtained could be written clearer. In fact I fail to see what the controls really bring since you would not expect the distributions of CI-edges to be the same between the two samples. It is only the kink in the real distribution for the real results that is the give away. 2) I think that the authors could have done many more and more interesting analyses with the unique data. However, I do not see this as mistake in the paper (the authors must decide what kind of paper they want to write - they chose the simple version). But in my view it is a bit of a lost opportunity.
--

	3) It is written "...We included all confidence intervals, but we anticipate the biases would be even more striking if we were able to restrict the sample to the primary outcomes of interest [18]..." I am unsure if this is right. I think I could argue for the reverse conclusion as well. The reason being that you often mention the primary outcome AND any surprising findings on the secondary in the abstract. This should suggest that there will be less publication bias in the primary outcome analyses.
--	--

VERSION 1 – AUTHOR RESPONSE

Charles Poole, University of North Carolina

General comments

1. The authors have done many research fields a good service by refining the examination of distributions of two-sided null p-values with specific attention to a bolus just below 0.05 (which I'll call the "barely significant bolus" BSB). A BSB can provide evidence of some, but not all, behaviors that impart selection biases to published literatures. Here, the authors extend this valuable work to BSBs of lower 95% confidence limits just above the null value and BSBs of upper 95% confidence limits just below the null value.

2. When authors bother to define the term "p-hacking," they do so in different ways. The authors give a rather narrow definition (p.3) that is the same as in reference 3: re-analyzing the same association in the same set of data in different ways in search of statistical significance (traditionally called "cooking"). Reference 4, however, gives a broader definition that includes not just cooking but varying the inclusion criteria for the analytic data set ("trimming") and extending the study in some way to collect more new data (e.g., by changing the "stopping rule"). Some authors might include even more behaviors under the "p-hacking" rubric, such as fraudulent fabrication or alteration of data ("fudging"). Any of these behaviors can produce a BSB. Hence, if the authors want to say that a BSB can identify or "detect p-hacking" (p.2 et seq.), I would recommend defining the term so that it includes every behavior that can produce a BSB and excludes every behavior that can't.

RESPONSE: Our thoughts were that "re-analyzing the data" encompassed practices such as alternative stopping rules or varying inclusion criteria, as they involve re-doing the analysis. We have added a clarifying sentence to respond to this comment and a similar comment from reviewer 3.

3. In previous work, the authors have argued persuasively that, though publication bias and p-hacking can produce an excess of statistically significant results in the published literature, only p-hacking can produce a BSB. In this paper, that distinction is not drawn quite as sharply or explicitly as it might be. I would encourage the authors to bring it out into the open and summarize the argument.

RESPONSE: It is challenging to pick apart how much of the BSB is due to p-hacking by reviewers and how much is selection by journals, or a combination of the two. We haven't gone into this distinction in detail here, because we focus on the simpler goal of highlighting the overall problem.

4. Experienced data analysts, among whom I would have to count myself, have reported anecdotally that, short of fudging, it's hard to make the null p-value change a great deal by the behaviors that can produce a BSB. If they are right, we might expect a shortfall in the p-value and confidence limit distributions in the region just barely on the statistically nonsignificant side of the divide. (If they have addressed this possibility in previous work, I apologize for not being familiar with it.)

RESPONSE: We cannot see this feature in our cumulative plots, but there may still be a shortfall that could be detected using statistical modelling. Our focus here is on the obvious excess of the BSB shown in the plots.

Specific comments

5. (Supplement) I don't think the STROBE checklist is appropriate here. It's for individual studies, not reviews.

RESPONSE: We found the STROBE checklist helpful, as our study does have a large observational component.

6. (p.3) Please give the specific page numbers on which reference 2, a book, gives or refers to empirical evidence on the frequency with which statistically significant results are perceived as being more important and valid. Of particular interest would be any empirical evidence, cited in reference 2 or elsewhere, showing that researchers equate statistical significance with negligible bias (i.e., with validity).

RESPONSE: Two relevant quotes from the Ziliak and McCloskey book are:

"By 1990 most subfields of epidemiology had, like economics and psychology, become predominantly Fisherian. Statistical significance came to mean 'epidemiological significance'. Statistical insignificance came to mean 'ignore the results'." (page 162 in book, which refers to a study by Savitz et al.)

"The automatic equating of statistically significant with clinically important, and non-significant with non-existent" (page 163 in book, referring to work by Altman).

As both these quotes appear in chapter 14, we've updated the citation to this specific chapter. We've also added the paper by Calin-Jageman and Cumming, which includes the quote, "Statistical significance [...] is treated as conclusive—worries about sample size or procedural error might go right out the door."

For confusing statistical significance with validity, a good reference is this randomised trial (our reference 1): "Testing for the presence of positive-outcome bias in peer review: a randomized controlled trial" Arch Intern Med. 2010 170(21):1934-9. doi: 10.1001/archinternmed.2010.406.

7. (p.3) Is carefully vetting which hypotheses to pursue, selecting the ones for which a study has the greatest statistical power and therefore gives the most precise estimates, a bad thing?

RESPONSE: We are concerned with researchers predominantly pursuing work that is more likely to be statistically significant, which is the tail wagging the dog. This would be a problem for scientific progress, because researchers might avoid potentially fruitful areas in favour of safer more publishable research.

8. (p.3) Please provide references that distinguish the role of journals from the roles of researchers, employers, and funding sources in publication bias, p-hacking, or both.

RESPONSE: We have added two papers from the recent special on p-values in The American Statistician.

9. (p.3) With regard to the assertion that journals "are also somewhat responsible," I wonder if the authors would address, or at least mention, the possibility of the following intergenerational conditioning effect: Suppose journals condition generation after generation of researchers to expect an uphill battle to get their statistically nonsignificant results published. Then the journals become enlightened, change their policies without fanfare, and quietly abandon their preference for statistically significant results. From that point on, it will misleadingly appear as though the fault lies with the researchers, whereas the underlying cause is still the past censorship by the journals.

RESPONSE: There has been more attention of p-hacking in statistical circles, but other researchers may hear a delayed message. For example, we were contacted in July this year by a group of clinical researchers to explain the NEJM editorial on p-values (DOI: 10.1056/NEJMe1906559). These researchers don't read papers in statistics journals, but they do pay attention to the NEJM.

The idea of a delayed change is an interesting idea, but we felt it was too speculative to include in the paper.

Reviewer: 2

Reviewer Name: Patrizio Tressoldi

This is a new contribution to the problem of publication bias consequent to the incentives to publish studies with only statistically significant results.

The authors clearly show that in the studies examined, there is a clear "confidence intervals - hacking".

I have some minor comments to add.

- both in the abstract and in the main text add the time intervals of the retrieved studies (from 1976 to ...). Furthermore, in the main text add the description of the search method;

RESPONSE: Agreed and added.

- on pag. 3, line 50, it could be added "...under a specified model, usually a null hypothesis testing, ...

RESPONSE: Agreed and added.

- in the Discussion, add a comment whether the observed CI - hacking may be prevalently due to errors in CI estimation or in intentional rounding or adjustments;

RESPONSE: It is very difficult to say. It could be that most hacking could be due to re-analysis. Given the uncertainty on this issue, we did not add to the discussion.

Reviewer: 3

Reviewer Name: Nicole Lazar, Professor

I'd like to see a clearer explanation of p-hacking (page 3, top). The authors write that "reanalysing data to get a p-value under 0.05" is known as p-hacking. This is a bit misleading, as one could infer that any secondary analyses are p-hacking. But prespecified secondary analysis would not be considered p-hacking typically. It should be made more clear that p-hacking occurs when the data are subjected to unplanned analysis (e.g. stratifying of units in the study according to their characteristics) in the search for "statistical significance" (typically $p < 0.05$).

RESPONSE: Agreed and added on page 3.

Likewise, the statements about the part that journals play could be clarified. Readers who are not as immersed in this literature and debate may not understand exactly the connection between the "statistically significant" and the "striking" result - elaboration here would be useful. The authors want to set the stage for the problem, but I find the discussion too general and likely to mislead readers who aren't already following the literature on this topic.

RESPONSE: We've clarified by defining striking as "with a very low probability of an observation being the result of chance".

I'd consider dropping the reference to the p-curve. It plays no further role in the work, and is not directly relevant. Instead, include references to papers that have investigated the excess of reported p-values close to the traditional cutoff points (0.01, 0.05, 0.1 are typical).

RESPONSE: Our paper and the p-curve papers are both try to use published results to uncover unbelieve patterns in inferential statistics. It would therefore feel an omission to not reference the previous work on the p-curve. We have included another paper that shows the excess of significant p-values in dentistry journals.

I don't think it's precisely correct to say that confidence intervals have been "advocated" as an "alternative to p-values" for presenting results, on account of avoiding bright-line thresholds. After all, a 95% CI gives a type of bright-line of its own (why not 90%? or 87%? etc.) Rather, I think the claim is that a confidence interval provides more information, for the same "cutoff value" about what values of the parameter are compatible with the data. It's a perhaps subtle distinction, but an important one. CIs can be abused just as easily, as this paper demonstrates!

RESPONSE: We have added a reference (DOI: 10.1080/00031305.2018.1518266) that specifically advocates for CIs in place of p-values (page 3). We do appreciate the reviewer's argument, which has also been made by Robinson, "If hypothesis testing and interval estimation are always mutually consistent then it seems to me that preferring interval estimates to hypothesis tests changes little" (DOI: 10.1080/00031305.2017.1415971). However, we do also think that confidence intervals are preferable to p-values, as argued by Calin-Jageman and Cumming (DOI: 10.1080/00031305.2018.1518266).

While it's no doubt true that researchers often misunderstand p-values, are confidence intervals any better understood? I'm not convinced and this does not therefore seem like a strong basis for studying confidence intervals in the way proposed. One could be more direct about it!

RESPONSE: In our experience we would say that the misunderstanding of p-values is the more serious problem because p-values have become the only game in town and the size of the difference (e.g., treatment effect) is very often ignored. Confidence intervals should be on a meaningful scale (although they are not immune to poor presentation, e.g., through poor scaling of the explanatory variable) and hence any inferential discussion is more likely to be based on the importance of the results. However, they are unlikely to be a panacea, and it is easy to find examples of recent papers where researchers have ignored the practical significance of their results despite using confidence intervals. The paper by Calin-Jageman and Cumming gives some good arguments for why confidence intervals should be better (DOI: 10.1080/00031305.2018.1518266).

In the Methods section, what is meant by "ratio confidence intervals"? (this is imprecise language) Ratio of what to what? It's made clearer in the next sentence but it might be better to lead with some context. Why are CIs of ratio estimates of particular interest? Presumably these are the most prevalent in the health research literature under study here? This should be clarified.

RESPONSE: We have moved the qualification of the ratios to the first sentence.

We used ratios because they can be extracted from the text more reliably as they are reported in standardised ways, e.g., using "odds ratio" or "OR". We have added this information to the methods (page 4).

Also it would be helpful for readers to provide more information about the text mining algorithm. The code alone may not be sufficient without some additional background. Or provide reference to a relevant supporting document. This may be in the Georgescu and Wren paper, but then what is the second text mining algorithm developed by the first author? There is not enough detail about this key aspect of the methods.

RESPONSE: Both sets of code are available on github for the interested reader (URLs included on page 4). Describing the text-mining algorithms would be very dry and space-consuming, as they are a long list of logical statements.

On page 5, under results, the phrasing about the upper interval limits just below 1 ("and so just inside the statistically significant threshold") is not clear.

RESPONSE: We have re-read this section and are happy with our wording. We note that the figure legend has some further explanation.

Regarding Figure S.3, the authors note that the steps in the ratios that are evident in the "zoomed in" plots are not simply due to rounding, since the discontinuity at 1 is clearly visible. But it does look like the steps at 1.1 and 1.2 (but, interestingly not at 0.9!) are of the same magnitude as the jump at 1. This is true in both the left and right panels (abstract and full-text, respectively), though somewhat more obvious in the former. Indeed the discontinuity at 1.1 for the intervals reported in the abstracts looks to be bigger than the jump at 1. I think this needs further exploration before one can claim that the cutoff of 1 has special behavior/bias.

RESPONSE: The main aim of these plots is to show that rounding is not the reason for the large change in the angle of the cumulative distribution. The plots show that the change in angle at the threshold of 1 is still clear even at this scale. The visible rounding is interesting and does seem to vary below and above 1, but this is of less interest to our overall idea.

Reviewer: 4

Reviewer Name: Paul Gustafson

I read this work with interest, and I am all around positively disposed to it. I have some comments and suggestions though.

p. 2: “are not statistically plausible” – could be misread as in indictment of statistical methods, when in fact the charge is levelled at research practice generally

RESPONSE: We have removed the word “statistically”. What we meant was that the results can be shown to be implausible when using a statistically summary of all results.

p. 3: Worth mentioning also funding agency practices and university promotion practice as incenting “significance seeking”?

RESPONSE: We are aware of work showing that promotion criteria are skewed towards “high impact” journals, which would then indirectly reward statistical significance. However, we are not aware of any funding or promotion criteria that specifically mention statistical significance.

The additional paper we have cited by Goodman outlines how funders could exert positive influence in this area.

p. 3, last para: little bit awkward to discuss correspondence of testing and interval estimation without some mention of “null” hypothesis.

RESPONSE: We have added a reference to null hypothesis testing.

p. 4: “mean was not within the CI” -> “point estimate was not within the CI”? (No real sense in which any of these estimates of *ratios” are means.)

RESPONSE: Changed as suggested.

p. 4: “highlights a change in the distribution” is, to my mind, awkward wording. A cdf is by definition an increasing function, so it is “changing” at every point along the abscissa.

RESPONSE: We have changed this to “useful for highlighting changes in the distribution”

p. 5: smooth..., with no discontinuities. To my mind (and to anyone interpreting with some level of math formality), “discontinuity” is not the right word. The cdf is continuous at $x=1$, but you are calling attention to a discontinuity in its derivative (the density function). I’m not sure what word is most appropriate, but a “bend” (versus not), would be a candidate.

RESPONSE: We have changed “discontinuity” to “jump”.

p. 6: “should be published regardless of their stat significance” ... to my mind, this is way too simplistic. This change, in isolation, would lead to journals as dumping grounds. Investigators would be incented to perform underpowered studies (even more so than is now the case), and it also punts on the question of going after novel/important/breakthrough hypotheses.

RESPONSE: We believe that statistical significance is a poor way to select what research gets published and is causing clear biases in the evidence-base. Publishing results regardless of the p-value is unlikely to solve all the issues in such a complex system, but it would increase the transparency around what research has been completed. As Calin-Jageman and Cumming state: “The file drawer problem is anathema to good science. It is a disgrace that this problem has persisted for so long since its initial recognition.” (DOI: 10.1080/00031305.2018.1518266).

I think the work could be better anchored to the existing literature. In particular, there aren’t citations to what are probably the two biggest recent “splashes” in this space – the special issue of Am. Stat. and the Nature commentary.

RESPONSE: We have added two papers from the recent special on p-values in The American Statistician.

Reviewer: 5

Reviewer Name: Theis Lange

The paper presents an interesting work on possible publication bias can be "seen" by looking at limits of confidence intervals. The short answer they find is: Yes it can.

By far the most impressive piece of work is the collection and coding of close to 1M confidence intervals from MEDLINE. The methods described for achieving this looks convincing, but it is of course impossible to review the precision of this part of the procedure. Based on this impressive data set the authors conduct some relative simple descriptive graphical analyses which indicates that there is a publication bias.

RESPONSE: The details can be checked by looking at the annotated code on github.

Comments:

1) The section on how control CIs are obtained could be written clearer. In fact I fail to see what the controls really bring since you would not expect the distributions of CI-edges to be the same between the two samples. It is only the kink in the real distribution for the real results that is the give away.

RESPONSE: Yes, we are not interested in the location of the distribution, but we are interested in its shape. We think the controls provide a very useful contrast to the results from pubmed.

2) I think that the authors could have done many more and more interesting analyses with the unique data. However, I do not see this as mistake in the paper (the authors must decide what kind of paper they want to write - they chose the simple version). But in my view it is a bit of a lost opportunity.

RESPONSE: The data are freely available on github for further analyses by any interested researcher.

3) It is written "...We included all confidence intervals, but we anticipate the biases would be even more striking if we were able to restrict the sample to the primary outcomes of interest [18]..." I am unsure if this is right. I think I could argue for the reverse conclusion as well. The reason being that you often mention the primary outcome AND any surprising findings on the secondary in the abstract. This should suggest that there will be less publication bias in the primary outcome analyses.

RESPONSE: This depends on the behaviour of researchers and how they try to grab attention. We might still expect researchers to present a primary outcome that is adjusted or not depending on which gave the best p-value (which is what citation [18] concerns). To be more cautious we have changed "we anticipate" to "it is possible".

VERSION 2 – REVIEW

REVIEWER	Charles Poole University of North Carolina USA
REVIEW RETURNED	18-Oct-2019

GENERAL COMMENTS	Please see the uploaded document.
-----------------------------------

REVIEWER	Nicole Lazar University of Georgia, USA
REVIEW RETURNED	15-Oct-2019

GENERAL COMMENTS	I have no additional comments.
--------------------------------

REVIEWER	Paul Gustafson University of British Columbia, Canada
REVIEW RETURNED	21-Oct-2019

GENERAL COMMENTS	I can live it, but as per my first review, again I'd draw attention to the wording around the behaviour of the distribution function. It's been changed from "discontinuity" to "jump," which is better in the sense that a very mathematical word is no longer being used incorrectly. However, it's still the case that "jump" best describes the behaviour of the corresponding density function, leading to a bend/kink in the distribution function.
---